# Influence of Fermions on Vortices in SU(2)-QCD

Zeinab Dehghan [1], Sedigheh Deldar [1], Manfried Faber [2,*], Rudolf Golubich [2] and Roman Höllwieser [3]

[1] Department of Physics, University of Tehran, Tehran 1439955961, Iran; zeinab.dehghan@ut.ac.ir (Z.D.); sdeldar@ut.ac.ir (S.D.)

[2] Atominstitut, Technische Universität Wien, 1040 Wien, Austria; rudolf.golubich@gmail.com

[3] Department of Physics, Bergische Universität Wuppertal, 42119 Wuppertal, Germany; roman.hoellwieser@gmail.com

* Correspondence: faber@kph.tuwien.ac.at

**Abstract:** Gauge fields control the dynamics of fermions, and, in addition, a back reaction of fermions on the gauge field is expected. This back reaction is investigated within the vortex picture of the QCD vacuum. We show that the center vortex model reproduces the string tension of the full theory also in the presence of fermionic fields.

**Keywords:** quantum chromodynamics; confinement; center vortex model; vacuum structure

**PACS:** 11.15.Ha; 12.38.Gc

## 1. Introduction

The QCD vacuum is highly nontrivial and has magnetic properties, as we have known since Savvidy's article [1]. The QCD vacuum should explain the non-perturbative properties of QCD, including confinement [2] and chiral symmetry breaking [3]. Lattice QCD puts the means at our disposal to answer the question about the important degrees of freedom of this non-perturbative vacuum. In the center vortex picture [4–6], the QCD vacuum is seen as a condensate of closed quantized magnetic flux tubes. These flux tubes have random shapes and evolve in time and therefore form closed surfaces in the dual space. They may expand and shrink, fuse and split and percolate in the confinement phase in all space-time directions and pierce Wilson loops randomly. Thus, Wilson loops asymptotically follow an exponential decay with the area. This is the area law of Wilson loops, which allows attributing the string tension to center vortices. The finite temperature phase transition is characterized by a loss of center symmetry and correspondingly by a loss of percolation in time direction. Therefore, vortices get static and only spatial Wilson loops keep showing the area law behavior.

Color electric charges are sources of electric flux according to Gauss's law. The electric flux between opposite color charges does not like to penetrate this magnetic "medium" of center vortices and shrinks to the well-known electric flux tube. On the other hand, the magnetic flux does not like to enter the electric string. Since fermions carry color charges, their dynamics is controlled by the gauge field. The presence of a fermion condensate is expected to suppress the quantized magnetic flux lines, and as a result the gluon condensate and therefore the string tension are reduced. Since, as usual, the lattice spacing is determined via the string tension, taking into account dynamical fermions leads to a decrease of the lattice spacing. In this article, we show a careful investigation of the string tension within the vortex picture of the QCD vacuum.

SU(2) and SU(3) QCD have equivalent non-perturbative properties. In a first study, we restrict our analysis to the simpler case of SU(2)-QCD. The most important difference between SU(2) and SU(3) QCD is the order of the finite temperature phase transition for a pure gluonic Lagrangian. There is a natural explanation for this difference from the structure of SU(2) and SU(3) vortices. There is only one non-trivial center element in the

group SU(2) and therefore one type of center vortices, whereas there are two non-trivial center elements for SU(3) and two types of vortices, allowing two vortices of the same type to fuse to the other type. This leads to a more stable structure of the net of vortices for SU(3) and to a first order phase transition, whereas in SU(2) the transition is of second order.

We investigate the fermionic back reaction on the gluonic degrees of freedom in SU(2) QCD. Visualizing the distribution of center vortices, this back reaction can be easily observed (see Figure 1). One can clearly see that dynamical fermions decrease the percolation of vortices. It is difficult to draw a closed surface in four dimensions. Therefore, we restrict ourselves to the three dimensional diagram of a time-slice and indicate the continuation to other slices by line stubs.

With fermions　　　　　　　　　　　　　　　　Without fermions

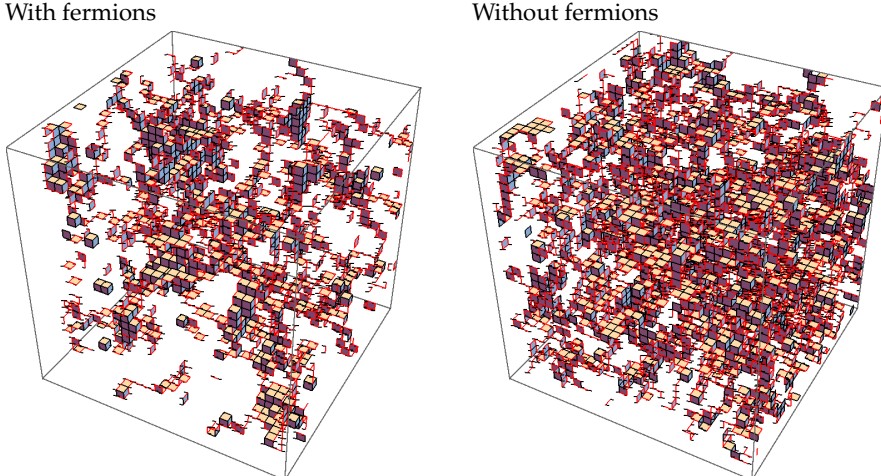

**Figure 1.** The closed vortex surface is visualized by showing the dual P-plaquettes of three-dimensional lattice slices. Stubs of red lines indicate plaquettes that are not fully part of the lattice slice shown. We clearly see that with fermions (**left**) an overall smaller amount of P-plaquettes is observed compared with the pure gluonic case (**right**). In both cases, one big vortex cluster dominates.

We want to quantify the effect in more detail. We are especially interested in the center vortex model [4–6] and its sensitivity to the fermionic back reaction. We also analyze the influence of fermionic fields on the geometric structure of the center vortex surface. This work compares four different estimates of the string tension, with and without fermions, in the full theory and in the vortex picture:

- via the potential calculated from the center degrees of freedom only, in pure gluonic ensembles;
- via the potential calculated from the center degrees of freedom only, in the presence of fermionic fields;
- via the potential in the full theory, in pure gluonic ensembles; and
- via the potential in the full theory, in the presence of fermionic fields.

With this comparison, we study the sensitivity of the center vortex model to the fermionic back reaction.

Our work is based on the QCD path integral which defines the vacuum to vacuum transition amplitude. In lattice QCD, we usually evaluate this amplitude on a lattice periodic in Euclidean time. Inserting a complete set of eigenstates of QCD with the quantum numbers of the vacuum in this amplitude results in an exponential decay of the eigenstates with the physical time extent $aN_t$ of the lattice, where $a$ is the lattice spacing and $N_t$ is the number of lattice sites in time direction. The inverse of this time extent therefore acts as a temperature of the ensemble. In a Monte-Carlo simulation, the states are occupied with the corresponding Boltzmann factor. The higher is the excitation, the smaller is the Boltzmann factor and the more difficult is the measurement of its properties. Finally, the excited states are vanishing in the noise.

The potential, as well as the string tension, can be calculated using *Wilson loops*

$$W(R, T) = \mathrm{Tr}\mathcal{P}\mathrm{e}^{\mathrm{i}\, g\, \int_{R \times T} A_\mu^a(x) t_a \mathrm{d}x^\mu}. \tag{1}$$

A loop of size $R \times T$ in space-time represents the world-line of a quark-antiquark system at distance $R$ propagating in the QCD vacuum for a time $T$. On a Euclidean lattice in SU(2)-QCD, a path ordered loop is determined by the product of link variables $U_\mu(x) \in$ SU(2) along the loop. Inserting a complete set of eigenstates of the quark-antiquark system into the expectation value $\langle W(R, T)\rangle$, the contributions of the eigenstates decay exponentially with Euclidean time $T$. The expectation values of Wilson loops can therefore be expanded in a series of eigenstates of the quark-antiquark system

$$\langle W(R, T)\rangle = \sum_{i=0}^{\infty} c_i \mathrm{e}^{-\varepsilon_i(R)T}, \tag{2}$$

For large times, $\langle W(R, T)\rangle$ is dominated by the ground state energy $\varepsilon_0(R)$. The more precise we determine $\lim_{T\to\infty}\langle W(R, T)\rangle$, the better is the precision of the quark-antiquark potential $V(R) := \varepsilon_0(R)$. Since the energy of the quark-antiquark system increases with the distance $R$, it follows from the above discussion that for increasing $R$ the signal for $V(R)$ is vanishing soon in the noise. How we handle this noise and how center vortices are detected is explained in Section 2. We assume that the potential is dominated by a Coulombic part at small $R$ but rises linearly for large $R$,

$$\varepsilon_0(R) = V(R) = V_0 + \sigma R - \frac{\alpha}{R}. \tag{3}$$

We use $\langle W(R, T)\rangle$ to approximate $\varepsilon_0(R)$, denoted as *1-exp fit*. $V_0$ parameterizes the scale dependent self-energy of the quark-antiquark sources. Wilson loops extracted from the center degrees of freedom are dominated by the long-range fluctuations of the QCD vacuum, hence we describe the potential within these degrees of freedom by

$$V_{\mathrm{CP}}(R) = v_0 + \sigma_{\mathrm{CP}}R. \tag{4}$$

The aim of this article is to investigate whether we can understand the string tension and its modification in the presence of fermions in the vortex model of confinement. Further, we present and discuss conceptual improvements to the gauge fixing procedure, required for the center vortex detection.

For systems with dynamical fermions one would expect string breaking when the energy of the system rises above twice the pion mass, but string breaking has been detected only using mesonic channels (see [7]). The center vortex model explains the asymptotic behavior of Wilson loops. There are indications that center vortices are sensitive to string breaking [8,9], but a direct measurement is not possible. From the vortex structure, we do not find any indication for string breaking which could show up as disintegration of the percolating vortex.

## 2. Materials and Methods

This section starts with a description of the parameters of the lattice configurations, used for our analysis. Then, our method of detecting center vortices with some novel improvements is discussed. We explain how the information about the geometric structure of the vortex surface can be acquired by smoothing procedures and we end with a detailed explanation of our method to extract the potential from Wilson loops. In each subsection, we list the intermediate results.

### 2.1. Simulation Specifications

We study the configurations described in [10] for chemical potential $\mu = 0$ with $S_G$ defined by a tree level improved Symanzik gauge action [11,12]

$$S_G = \beta \left( c_0 \sum_{\square}(1 - \frac{1}{2}\text{Tr}\,\square) + c_1 \sum_{\square\square}(1 - \frac{1}{2}\text{Tr}\,\square\square) \right), \tag{5}$$

with coefficients $c_0 = 5/3$ and $c_1 = -1/12$. The first sum corresponds to the Wilson action with $\square$ indicating single unoriented plaquettes, while the second sum uses rectangular Wilson loops built of 6 links, symbolized by $\square\square$. The inverse coupling is defined as $\beta = \frac{4}{g^2}$ for SU(2).

For the fermionic degrees of freedom, staggered fermions are used with an action of the form

$$S_F = \sum_{x,y} \bar{\psi}_x M(m)_{x,y} \psi_y + \frac{\lambda}{2} \sum_x \left( \psi_x^T \tau_2 \psi_x + \bar{\psi}_x \tau_2 \bar{\psi}_x^T \right), \tag{6}$$

with $\tau_i$ being the Pauli matrices and

$$M(m)_{xy} = m\delta_{xy} + \frac{1}{2}\sum_{\nu=1}^{4} \eta_\nu(x)\left[ U_{x,\nu}\delta_{x+h_\nu,y} - U^\dagger_{x-h_\nu,\nu}\delta_{x-h_\nu,y} \right], \tag{7}$$

where $\bar{\psi}$, $\psi$ are staggered fermion fields, $a$ is the lattice spacing, $m$ is the bare quark mass, $U_{x,\nu}$ is a SU(2) element corresponding to a link at position $x$ is in direction and $\mu$ and $\eta_\nu(x)$ are the standard staggered phase factors: $\eta_1(x) = 1$, $\eta_\nu(x) = (-1)^{x_1+\dots+x_{\nu-1}}, \nu = 2, 3, 4$. The total action is given by $S = S_G + S_F$. Integrating out the fermionic degrees of freedom, the partition function with $N_f = 2$ is given by

$$Z = \int DU \, e^{-S_G} \, (\det(M^\dagger M) + \lambda^2)^{\frac{1}{4}}. \tag{8}$$

The properties of 1000 configurations of size $32^4$ with $\beta = 1.8$, quark mass parameter $m = 0.0075$ (corresponding to $m_\pi = 740(40)$ MeV with lattice spacing $a = 0.044$ fm), and $\lambda = 0.00075$ are compared to 1000 pure gluonic configurations at the same inverse coupling $\beta$. For both sets of 1000 configurations, we extract the potentials from all available Wilson loop data and compare them with the string tensions resulting from the two sets of $40 \times 100$ center projected configurations. In this way, we try to answer the question, if in the presence of dynamical fermions the center degrees of freedom determine the string tension of the gluonic flux tube in quark–antiquark systems.

### 2.2. Center Vortex Detection

Assuming that center excitations are the relevant degrees of freedom for confinement, we detect these center vortices within the lattice configurations. We first identify gauge matrices $\Omega(x) \in$ SU(2) at each site $x^\mu$ maximizing the functional

$$R_F = \sum_x \sum_\mu | \text{Tr}[\acute{U}_\mu(x)] |^2 \quad \text{with} \quad \acute{U}_\mu(x) = \Omega(x + e_\mu)U_\mu(x)\Omega^\dagger(x). \tag{9}$$

After fixing the gauge, the link variables $\acute{U}_\mu(x)$ are projected on the center degrees of freedom, that is $\pm 1$ for SU(2), to neglect short range properties and keep only long-range effects

$$U_\mu(x) \to Z_\mu(x) \equiv \text{signTr}[U_\mu(x)]. \tag{10}$$

After performing the center projection, the center projected plaquettes resulting from the vortex detection are the products of four center elements. The projected plaquettes are non-trivial, known as P-plaquettes, $U_\square = -1$, if one or three links are non-trivial. In the four-dimensional lattice, a given link belongs to six plaquettes. On the dual lattice,

the corresponding six plaquettes build the surface of a cube. Therefore, the duals of P-plaquettes form closed surfaces, dual P-vortices which correspond to the closed flux line evolving in time.

This procedure is the original DMCG [13] in which a gradient climb with *over relaxation* was used to maximize the gauge functional. From a few gauge copies only, produced in this way, the one with the highest value of the functional is usually chosen for further analysis. This method leads to promising results, but improvements at maximizing the gauge functional using *simulated annealing* have brought a flaw to light—the many local maxima of $R_F$ do not necessarily correspond to the same physics. Bornyakov et al. [14] showed that there exist local maxima of the gauge functional that underestimate the string tension. We have been able to resolve these problems for smaller lattices using improved version of the gauge fixing routines based on non-trivial center regions [15–17], but our implementation was not able to handle the big lattices used in this work. Taking a closer look at the problem at hand, we can look for a different approach. We now consider Creutz ratios to estimate the string tension

$$\sigma \approx \chi(R) = -\ln \frac{\langle W(R+1, R+1)\rangle \, \langle W(R, R)\rangle}{\langle W(R, R+1)\rangle \, \langle W(R+1, R)\rangle},$$

(11)

with Wilson loops $W(R, R)$ of size $R = T$. Some probability densities for the relation between the values of the gauge functional $R_F$ and the Creutz ratio $\chi(R)$ for individual configurations are shown in Figure 2. This determination is based on 40 configurations with 100 gauge copies for configurations with (left) and without (right) dynamical fermions. For Creutz ratios of small Wilson loops, we observe a nearly linear relation between the two quantities reflecting the finding of Bornyakov et al. [14]: there exist gauge copies of the configurations with maximal $R_F$ and very low $\sigma$. With increasing size of Wilson loops, this correlation weakens. Nevertheless, the request to maximize the gauge functional (9) fails.

Another observation is of high interest: extremely small and large values of the gauge functional are strongly suppressed in the probability densities. Instead of looking for higher local maxima of the gauge functional, we propose a different approach: "ensemble averaged maximal center gauge" (EaMCG). We produce many random gauge copies, approach the next local maximum by the gradient method and take the average of the ensemble. The idea is that not the best local maximum alone carries the physical meaning, but the average over all local maxima does: maxima with a higher value of the gauge functional result in a reduced string tension, but they are not dominating the ensemble. The same holds for lower valued maxima, possibly overestimating the string tension.

Taking again a look at Figure 2, it can be seen that the average values and the most probable values are in good agreement for small loops. This is shown in more detail in Figure 3 for Creutz ratios of different loop-sizes. The fact that differences increase with loop sizes can probably be explained by the lack of statistics for the Creutz ratios of single configurations. Until the values start to deviate from one another, there is a variation of 10% over the whole $R$-region. Despite the low statistics of a single configuration, the intermediate loop sizes already reproduce the asymptotic behavior and let us expect the possibility for a more precise determination. First averaging over Wilson loops and then calculating Creutz ratios gives much more stable results (see $\chi_W(R)$ in Figure 3). The final estimate of the string tensions in Sections 2.4 and 3 is based on the determination of the potential.

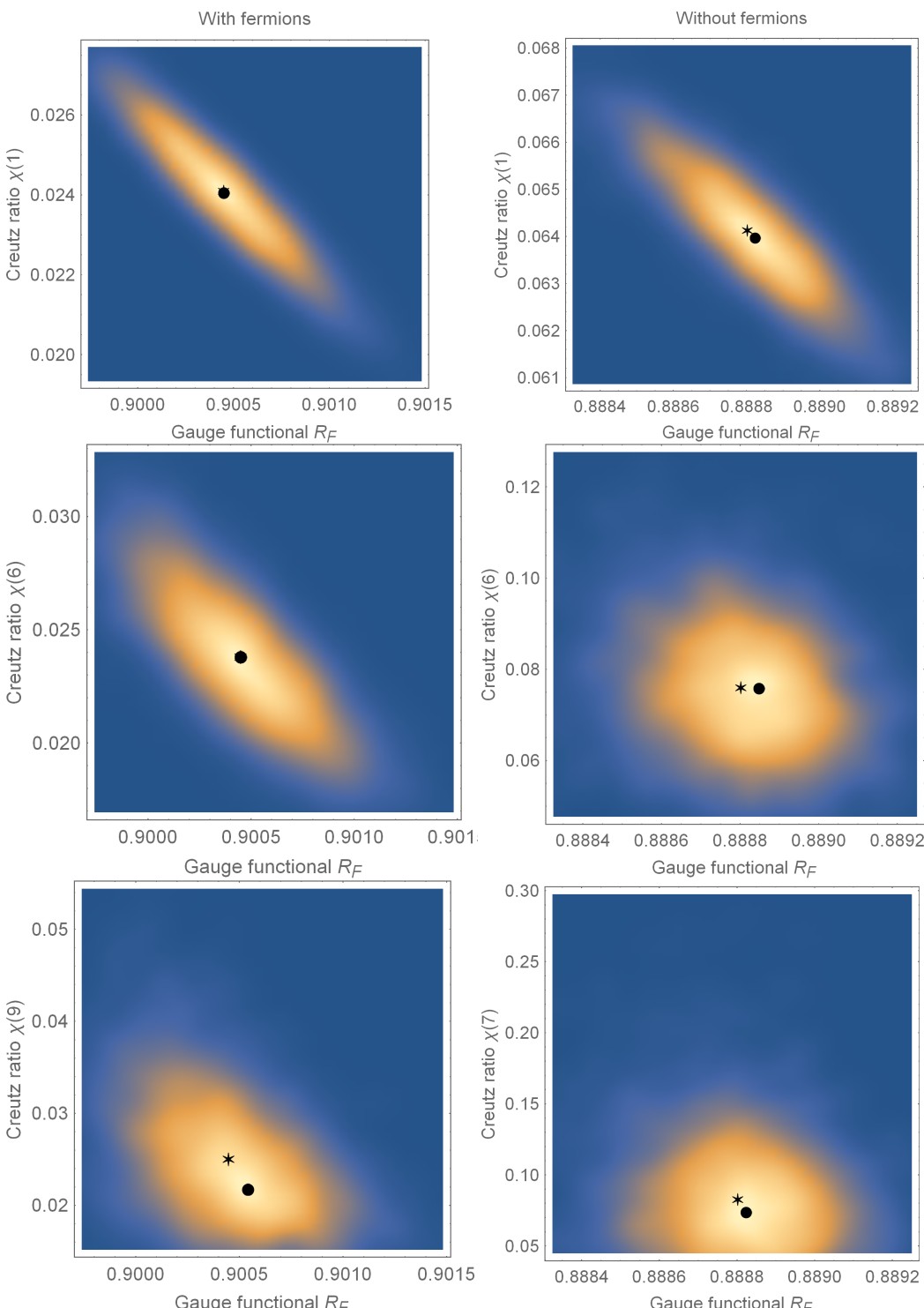

**Figure 2.** These probability densities specify the relation between the values of gauge functional and Creutz ratio for individual configurations. This determination is based on 40 configurations with 100 gauge copies for the configurations with dynamical fermions (**left**) and without (**right**). For Creutz ratios of small Wilson loops, we observe a nearly linear relation. With increasing size of the Wilson loops this correlation weakens. We marked the average values (star) and the most probable values (circle) of the distributions.

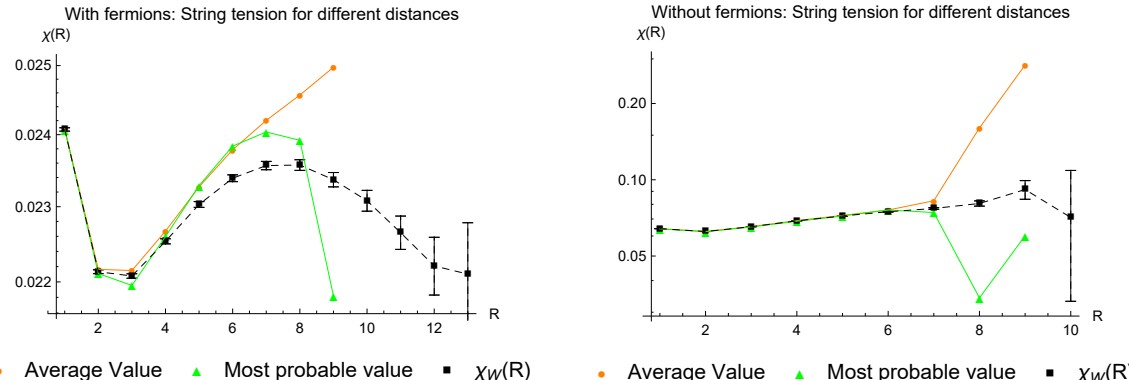

**Figure 3.** The average and most probable values of $\chi(R)$ are compared for simulations with and without fermions. This complements the probability densities of Figure 2. The increasing discrepancy between the two quantities with larger *R* can probably be explained by the low precision of Creutz ratios of single configurations. Until the values start to deviate from one another, the variations of $\chi(R)$ are of the order of 10%. For comparison, we show also the more precise Creutz ratios $\chi_W(R)$ extracted from averages of Wilson loops.

Thus far, we have calculated the Creutz ratios for single configurations of the ensemble and have taken the average afterwards. The EaMCG itself does not average over Creutz ratios, but combines first the Wilson loops of all gauge copies and configurations. From this fact, it is possible to extract the quark anti-quark potential, which allows a more precise determination of the string tension from the center vortex model.

In the respective single configurations, we observe one percolating large cluster that is surrounded and traversed by small fluctuations. These result in an increased number of P-plaquettes that do not contribute to the string tension. Analyzing these distortions, we gain insight on the influence of fermions on the geometric structure of the vortex surface.

### 2.3. Smoothing the Vortex Surface

There exist several procedures for smoothing the vortex surface by removing distortions. These procedures are discussed in detail in [18]. They do not modify the long range effects of the configuration. To get information about the smoothness of the vortex surface with and without fermions, we use the smoothing steps depicted in Figure 4. The smoothing 0 is not depicted, which removes unit-cubes.

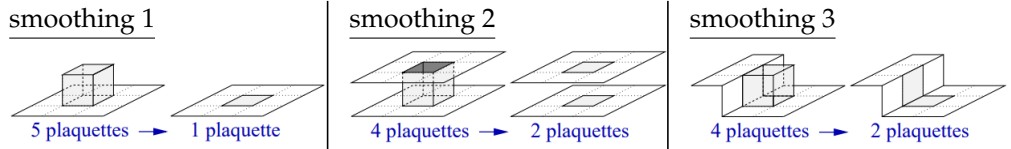

**Figure 4.** The effect of the smoothing procedures on the vortex surface is depicted, taken from ([19] Figure 5.8). We distinguish *warts* (**left**), *bottlenecks* (**middle**) and *stumbling blocks* (**right**). The *unit cubes* are not depicted, which are simply deleted.

The smoothing steps 1–3 cut out parts of the vortex surface and closes the emerging holes with a flat surface. In this way, short-range fluctuations of the vortex surface are suppressed. We first count the P-plaquettes without any smoothing performed, and then the loss of P-plaquettes for the respective smoothing steps is determined. The results are given in Table 1.

**Table 1.** Reduction of the total count of P-plaquettes for different smoothing procedures.

| P-plaquette Reduction | smoothing 0 | smoothing 1 | smoothing 2 | smoothing 3 |
|---|---|---|---|---|
| With fermions | 12.5% | 10.1% | 24% | 10.2% |
| Without fermions | 7% | 10.6% | 27.8% | 10.9% |

This quantifies the percentage of the respective structures depicted in Figure 4. When fermions are present, we clearly have a higher proportion of unit cubes and a lower proportion of bottlenecks than without fermions.

By restricting this analysis to the single percolating vortex cluster, we gain information about the long range excitations. The results are given in Table 2. The reduction in the proportion of bottlenecks is also seen here. The presence of fermions leads to a smoother surface of the percolating cluster.

**Table 2.** Reduction of P-plaquettes for the percolating vortex cluster for different smoothing procedures.

| Reduction within Cluster | smoothing 1 | smoothing 2 | smoothing 3 |
|---|---|---|---|
| With fermions | 8.6% | 24.5% | 8.8% |
| Without fermions | 9.6% | 28.1% | 10% |

*2.4. Potential Fits and Noise Handling*

When extracting the potential from Wilson loops, two effects have to be taken care of:

- for small areas, the loop averages are influenced by short range fluctuations; and
- with increasing area, the data suffer from statistical noise and soon the errors get larger than the signal.

An example for a 1-exponential fit to Wilson loops $\langle W(R, T) \rangle$ for given $R$ and $T \geq T_i$ (see Equation (2)) is shown in the left diagram of Figure 5. The dependence of this example on the initial $T = T_i$ is depicted in the right diagram.

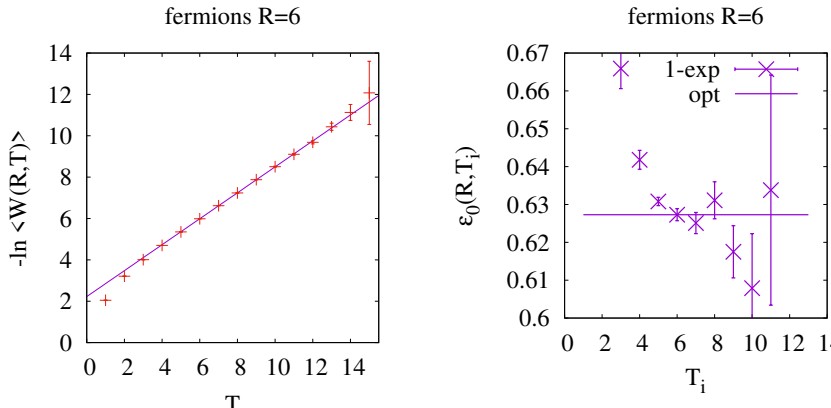

**Figure 5.** (**Left**) Example of the optimal 1-exponential fit of Wilson loops for given $R$. (**Right**) Dependence of $\varepsilon_0(R, T_i)$ on the fit region $T \geq T_i$. The line marks the fit for the optimal value for $T_i$.

At lower $T_i$, an increase of $T_i$ causes large changes of the fit parameters, but with growing $T_i$ these changes become smaller until a most stationary point is reached which may be hidden behind a strong increase of error bars. With the naked eye, one sees data that result in quite good fits, but finding analytic or numeric criteria for the choice of $T_i$ proves difficult. The smaller is the change of the values of the fit parameters, the smaller are the error bars of Wilson loops, and a rapid increase of the *p*-value of the fits often coincide,

but this is not a general rule. Our criteria to choose $T_i$ is based on identifying the first local minimum of an error quantifier

$$\text{Err} := \frac{2}{3}\langle\Delta_{\delta i}\rangle + \frac{1}{3}\langle\Delta_{\text{err}}\rangle. \tag{12}$$

Here, $\langle\Delta_{\delta i}\rangle$ denotes the average change of the fit parameter $\varepsilon(R, T_{i\pm 1})$ when decreasing or increasing $T_i$; and $\langle\Delta_{\text{err}}\rangle$ denotes the average over the error bars of $\varepsilon(R, T_{i-1})$, $\varepsilon(R, T_i)$, and $\varepsilon(R, T_{i+1})$. The weight factors are chosen to avoid the choice of occasionally nearly stationary regions with large error bars. For $R > 3$, we prevent any further increase of $T_i$, because with increasing $R$ the error bars start to grow earlier. The example in Figure 5 tries to convince that the selection of $T_i$ based on the error quantifier results in optimal fits under the boundary conditions of systematic deviations for low $T_i$ and increasing error bars for high $T_i$. Using this procedure, we determine the potential for the whole range of $R$-values, which allows extracting the slope of the potential at large values of $R$.

## 3. Summarized Results and Discussion

The fermionic back reaction on the string tension is clearly observed in the full theory as well as for EaMCG (Ensemble averaged Maximal Center Gauge), where the link variables of the gauge field are projected to $Z_2$. The potentials for the gluonic and fermionic configurations are depicted in Figure 6 and compare the full SU(2) theory with the $Z_2$ theory. The string tension was extracted by fitting the respective Equation (3) or (4) to the data describing the potential. The resulting parameters of these fits are given in Table 3. The relevant parameters to compare are $\sigma$ and $\sigma_{\text{CP}}$.

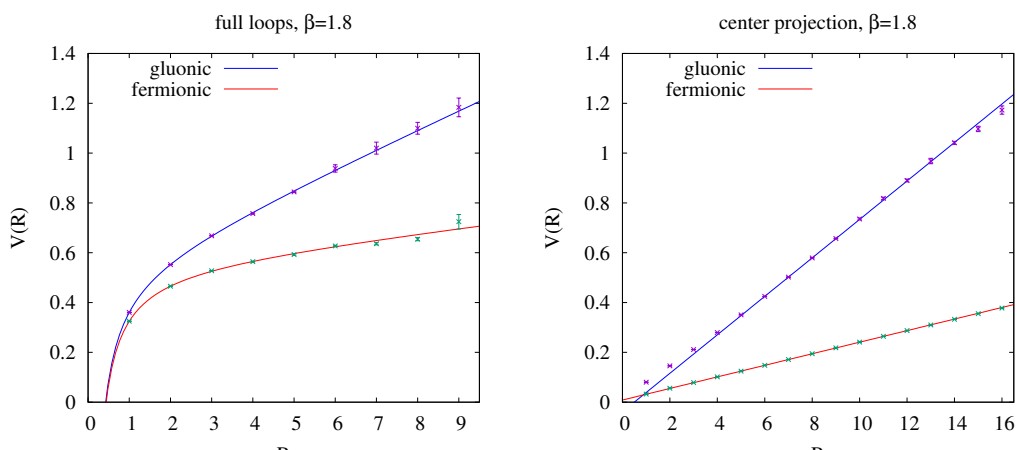

**Figure 6.** (**Left**) Potential $V(R)$ in lattice units between two sources in the fundamental representation. There is a large difference between the string tensions for pure gluonic configurations ("gluonic") and in the presence of one species of dynamical fermions. (**Right**) Potentials extracted from Wilson loops after ensemble averaged maximal center projection are depicted for pure gluonic configurations and for configurations with dynamical fermions. Due to the removal of short range fluctuations the potentials are in both cases almost linearly increasing with the lattice distance $R$. Data are fitted by linear functions. For gluonic (fermionic) configurations, only data with $R \geq 6(2)$ are fitted.

Without fermions, both estimates for $\sigma$ are compatible within errors to one another: In the full $SU(2)$, we observe $\sigma = 0.0756(12)$ compared to $\sigma_{\text{CP}} = 0.07691(13)$ in the $Z_2$ description. With fermions the full SU(2) theory results with $\sigma = 0.0199(9)$ a lower value than the $Z_2$ theory with $\sigma_{\text{CP}} = 0.02291(5)$. In all cases, we clearly observe that the presence of fermions reduces the string tension in lattice units: The back reaction is observed in the full SU(2) theory and also reproduced by the center vortices. The determination of the lattice spacing via the usual formula ($a = \sqrt{\chi}/2.23$ fm, corresponding

to $\chi = (440 \text{ MeV})^2$) results in 0.123(1) fm for the gluonic configurations and 0.0633(15) fm for the fermionic configurations.

**Table 3.** The parameters for the fits according Equations (3) and (4) in Figure 6 allow a direct comparison of the respective string tensions. A strong suppression of the Coulomb part can be seen in the $Z_2$ theory.

| Theory | SU(2) | | | $Z_2$ | |
|---|---|---|---|---|---|
| Parameter | $V_0$ | $\sigma$ | $\alpha$ | $v_0$ | $\sigma_{CP}$ |
| gluonic | 0.5175(38) | 0.0756(12) | 0.2326(26) | $-0.0366(8)$ | 0.07691(13) |
| fermionic | 0.5464(27) | 0.0199(9) | 0.2414(19) | 0.01027(13) | 0.02291(5) |

Concerning the geometric structure of the vortex surface, we observe that the presence of fermions increases the number of isolated short range fluctuations (see Table 1): without fermions, about 6.98% of the P-plaquettes are part of isolated unit cubes, whereas, with fermions, this proportion increases to 12.45%. The proportion of P-plaquettes belonging to bottlenecks is in total decreased from 27.81% to 24%. Fermions increase the amount of unit cubes, but decrease the amount of bottlenecks.

Restricting the analysis to the long-ranged cluster we observe a decrease of fluctuations, especially bottlenecks, when fermions are present (see Table 2): the proportion of P-plaquettes belonging to bottlenecks is reduced from 28.12% to 24.45%. All other fluctuations are only reduced by about 1%.

From this, we can conclude that the presence of fermions causes short range fluctuations to detach from the vortex surface, resulting in a more smooth vortex surface that is surrounded by an increased number of isolated short range fluctuations.

**Author Contributions:** Conceptualization, M.F. and R.H.; methodology, Z.D., M.F., R.H. and R.G.; validation, S.D.; investigation, Z.D. and R.G.; data curation, Z.D. and M.F.; writing—original draft preparation, Z.D. and R.G.; writing—review and editing, Z.D. and S.D.; visualization, R.G. and R.H. All authors have read and agreed to the published version of the manuscript.

**Funding:** This research received no external funding.

**Data Availability Statement:** The data presented in this study are available on request from the corresponding author.

**Acknowledgments:** We thank Aleksandr Nikolaev, Nikita Astrakhantsev and Andrey Kotov for their cooperation in the early stage of this investigation and Vitaly Bornyakov for important advice.

**Conflicts of Interest:** The authors declare no conflict of interest.

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
