# Peer review of "Influence of Fermions on Vortices in SU(2)-QCD"

_universe, doi:10.3390/universe7050130_

Round 1

Reviewer 1 Report

In this manuscript, the authors analyze the response of the SU(2) QCD
vacuum in the presence of static color sources created by a Wilson loop. In
particular, they analyze whether the center vortex picture is able to capture
the potential between the static charges. For completeness, they also apply
their methods to the pure SU(2) Yang-Mills case, where the string tension
is known to be described by the center-vortex ensemble. The simulations
show that dynamical quarks lead to a smaller number of P-plaquettes when
compared with the pure gluonic case, which in turn reduces the observed
string tension. As a main result, not only the string tension in pure Yang-
Mills is reproduced (as expected) but also the full lattice string tension in the
presence of dynamical quarks is reproduced by the center-vortex ensemble.
I consider that this work is interesting and suitable to be published in
'Universe'. With regard to the presentation, the procedure to  fix the dynamical quark mass parameter (m) could be explained with further details. Correspondingly, when dynamical quarks are present, a brief discussion about why a confining potential between static charges is preferred to a string-breaking regime would be welcome. Finally, in the introduction, there are some typos to be corrected and the writing could be improved.

Author Response

Dear Reviewer,

Thank you for your comments.

Concerning the dynamical quark mass parameter (m), we have made minor changes after Eq. (8):

"The properties of 1000 configurations of size $32^4$ with $\beta=1.8$, bare quark mass $m=0.0075$ (corresponding to $m_\pi= 740(40) MeV$) and $\lambda=0.00075$ are compared to 1000 pure gluonic configurations at the same inverse coupling $\beta$."

Indeed, we had forgotten to write the value for m - thank you for the reminder!

We added an explanation why a confining potential between static charges is preferred to a string-breaking regime at the end of the introduction: 

"For systems with dynamical fermions one would expect string breaking when the energy of the system rises above twice the pion mass, but string breaking has been detected only using mesonic channels, see Ref.}~\cite{Bulava:2019iut}\hl{. The center vortex model explains the asymptotic behaviour of Wilson loops. There are indications that center vortices are sensitive to string breaking} \cite{Hollwieser:2015qea, Altarawneh:2015bya}\hl{, but a direct measurement was not possible. From the vortex structure we did not find any indication for string breaking which could show up as disintegration of the percolating vortex."

We also found the typos you mentioned. They are corrected now and we will see whether we can improve the wording or not.

All modifications are highlighted.

With best regards
from the authors

Reviewer 2 Report

The QCD vacuum is non-trivial and reveals non-perturbative aspects, one of which is the vortices. In SU(2) this character can be seen clearly. 
I would like to recommend this manuscript to be published. This is because the careful analyses, especially, of the influence of the dynamical fermions. The vortex surface smoothing and the noise handling are instructive.  Results in Figure 3 are very interesting.

The followings are suggestions that may make the paper more clear for readers.

1) As the authors know, more papers were published on this topic by European, US, and Russian groups. Such references can help the readers to know the situation of the research.

2) After Eq.(3), the authors fit the result with 1-exp fit.  Is it possible to provide the contribution of the higher states quantitatively (c1, c2, and epsion1, epsilon2) ?

3) After Eq.(4), the authors give m_pi in the physical unit. If the authors give the value of a (lattice spacing), it is a useful information.

4) After Eq.(4), the authors try to ï½—eaken the Gribov copy effects by 100 random gauge copies. The effect of the Gribov copy was studied, but for Direct Maximam Center Gauge, not all readers know 100 is enough or not.  It is nice if the authors can show for example 100 and 200 give the same reults.

5) In the 3rd paragraph on page 1: this differences --> these differences

Author Response

Dear Reviewer,

Thank you for your comments.

1) Concerning other publications on this topic: We searched again but did not find any literature on center vortices in combination with fermions. If you know some relevant publications, we would happily include them in the references.

2) We tried intensively to produce 2-exp and 3-exp fits, but the results are not reliable enough for publication.

3) We have added the values of the lattice spacing after Table 3.

4) Each gauge copy takes about 2000 iterations before convergence is observed. The calculation of 100 gauge copies is already very involved for our lattices of size $32^4$. The calculation of further 100 gauge copies for the 40 configurations would delay the publications by several weeks. Fig. 2 shows that the density plots do not show irregularities (for example, multiple maxima). Therefore, we do not expect any relevant changes of the results with an increased number of gauge copies. In fact, we reproduced similar results with 100 gauge copies of only 10 configurations. The results for the center projected string tension are anyway very stable. The calculations in SU(2) would benefit from an increased number of configurations.

5) The mistake is corrected. Thank you!

All modifications are highlighted.

With best regards
from the authors